# VPO: Reasoning Preferences Optimization Based on $\mathcal{V}$-Usable Information

Zecheng Wang[1,2*], Chunshan Li[1✉], Yupeng Zhang[2*], Han Liu[3], Bingning Wang[2✉], Dianhui Chu[1], and Dianbo Sui[1✉]

[1]*Harbin Institute of Technology*
[2]*WeChat, Tencent Inc*
[3]*Tsinghua University*

zechengwang@stu.hit.edu.cn
{lics,chudh,suidianbo}@hit.edu.cn
{xfelixzhang,danialwang}@tencent.com,
han-liu18@mails.tsinghua.edu.cn

## Abstract

Direct Preference Optimization (DPO) is a widely used preference optimization algorithm in large language model (LLM) alignment, which reparameterizes the reward function in reinforcement learning with human feedback (RLHF) without requiring a separate reward model. However, during the DPO training process, when a large negative gradient is applied to low-confidence samples, LLMs with a softmax output head tend to squeeze the confidence in the model's output distribution towards the highest-confidence sentence, which may lead to a decrease in the confidence of both preference and non-preference samples, while increasing the confidence of unrelated tokens. This phenomenon becomes more complex in reasoning tasks. In this work, focusing on reasoning tasks, we propose VPO, a negative gradient constraint method for human non-preference samples based on $\mathcal{V}$-usable information. By using $\mathcal{V}$-usable information to measure the similarity between preference pairs and selectively constrain the negative gradient, VPO can alleviate the squeezing effect of DPO, enhance alignment with the generation objective, and maintain the model's ability to distinguish between preference and non-preference samples. We compare VPO with DPO and its latest variants on mathematical reasoning tasks using the LLama 3.1 and Qwen 2.5 series, including both Base and Instruct models. Our results demonstrate that VPO consistently and significantly outperforms existing methods. Specifically, on Qwen2.5-7B-Base, VPO achieves 7.80% and 13.25% improvement over DPO on MATH500 and AMC23, respectively. We also conduct ablation experiments and in-depth analysis on VPO to explain its effectiveness and rationale.

## 1 Introduction

Recently, large language models (LLMs) have undergone rapid iteration and evolution, attracting widespread attention across various industries, such as finance, healthcare, and education [9, 26, 4, 38, 31]. Post-training has emerged as a critical phase in the LLM training pipeline, serving to align model outputs with human values and preferences, ensuring their safety and impartiality, and enhancing reasoning reliability. Preference optimization, as a post-training alignment method, has been shown to be more advantageous than simple supervised fine-tuning in aligning with human preferences.

---

This work was supported by Tencent Inc, China.
*Equal contribution. ✉Corresponding authors.

Compared to the multi-stage training process of reinforcement learning from human feedback (RLHF) [27, 36, 6], which first requires training a reward model from preference data and then optimizing the policy model to maximize this reward, Direct preference optimization (DPO) [32] directly derives the reward signal from preference data by reparameterizing RLHF's reward function, thus bypassing the need for an explicit reward model. However, DPO's implicit reward is calculated based on the log probability ratio between the policy model and the reference model. Since the reference model is absent during inference, there exists a discrepancy between DPO's optimization objective and the goal of optimizing the log probability of preference samples in inference. Besides, DPO is sensitive to the initial policy model and often exhibits the phenomenon where log-probabilities decrease simultaneously for both preference and non-preference samples [11].

Recently, "squeezing effect" [33] is introduced to explain the underlying cause of the decrease in the log probability of preference data during the DPO training process. Specifically, for any LLM with a softmax output head, when a sample's confidence resides in the "valley" region of the model's predicted distribution, applying a large negative gradient to that sample will significantly suppress the entire output distribution curve, except for the sentence with the highest confidence before the update. Since DPO uses fixed, pre-collected "off-policy" data, there is a distribution shift between the updated policy model and the initial policy model used for data collection, leading to a non-uniform confidence distribution over preference pairs. Besides, DPO focuses more on how to avoid generating non-preference samples [11], which makes non-preference samples more likely to fall into the "valley" region under the influence of the negative gradient. When the token with the highest confidence is unrelated to the preference pair, it causes the log probabilities of both preference and non-preference samples to decrease simultaneously. The issue of preference sample probability decline in DPO becomes more severe in reasoning tasks, even leading to a decrease in model performance [28, 5, 22, 29].

In this work, to alleviate this issue, we propose VPO, a negative gradient constraint method for human non-preference samples based on $\mathcal{V}$-usable information. $\mathcal{V}$-usable information is proposed for quantifying the amount of information about the label $Y$ that can be extracted from the input $X$ through a given model family $\mathcal{V}$ [10, 42]. A series of studies have extended $\mathcal{V}$-usable information to quantify the amount of label-related information introduced by the chain-of-thought (CoT) generated by LLMs, relative to the input $X$ [30, 39]. VPO consists of two main components: (1) Non-preference sample information measurement, which uses $\mathcal{V}$-usable information to measure the additional amount of information about the label $Y$ contained in the non-preference sample reasoning chain, relative to the input $X$. (2) Negative gradient constraint, which constrains the negative gradients of non-preference samples based on their $\mathcal{V}$-usable information.

In our methodology, we first demonstrate that the negative gradient constraint in DPO can enhance the positive gradient of preference samples. Therefore, this constraint helps restrain the squeezing effect caused by large negative gradients, while also directing preference samples toward the region of highest confidence. Next, we categorize non-preference samples into two types based on the $\mathcal{V}$-usable information in their reasoning chains: (1) Samples that introduce positive $\mathcal{V}$-usable information about the label. These samples are considered highly correlated with preference samples, since nearly all $\mathcal{V}$-usable information from the reasoning chains of preference samples is positive. For such samples, we impose no constraints to ensure the model can effectively distinguish them. (2) Samples that introduce negative $\mathcal{V}$-usable information about the label. These samples have a weaker correlation with preference samples, thus the positive gradient from preference samples has a weaker impact on them. As a result, they experience a relatively larger negative gradient, causing their confidence to drop sharply, which makes them more prone to the squeezing effect. For these samples, we apply a stronger negative gradient constraint based on their normalized $\mathcal{V}$-usable information. In conclusion, by applying selective negative gradient constraints to different non-preference samples, VPO mitigates the squeezing effect caused by large negative gradients, enhances the log probability of preference samples, and prevents the issue of a small log probability gap between preference samples due to excessive constraints on negative gradients.

Extensive experiments demonstrate that, on both the base and instruct models of the Qwen-2.5 [31] and LLaMA 3.1 [14] series, VPO consistently exhibits superior overall performance on reasoning tasks compared to DPO and its variants. For instance, on the Qwen-2.5-7B-Base model, VPO outperforms DPO by 7.80%, 4.02%, 2.57%, 3.41%, and 12.25% on MATH500 [17], GSM8k [8], Minerva MATH, [20] Olympiad MATH[16], and AMC23, respectively. In addition, when DPO encounters a squeezing effect that causes a decline in model performance, VPO is still able to improve

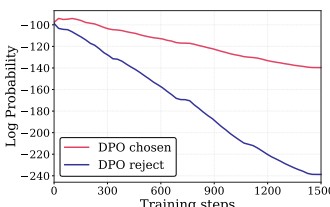 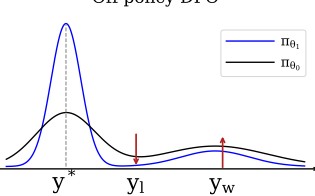 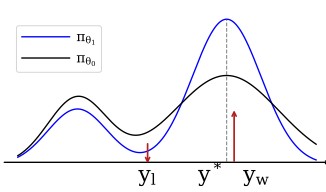

Off-policy DPO VPO

(a) Log-Likelihood decline of pref-
erence samples in DPO

(b) The squeezing effect in DPO

(c) Optimization performance of
VPO

Figure 1: Sample confidence changes in DPO. (a) DPO reduces log-likelihood for both preference/non-preference samples. (b) Squeezing effect in DPO from strong negative gradients applied to low-confidence non-preference samples. (c) VPO's gradient constraints restrain the squeezing effect.

model performance. Through ablation experiments and in-depth analysis of VPO, we further reveal the effectiveness and rationale behind its key design choices.

## 2    VPO: Reasoning Preference Optimization Based on $\mathcal{V}$-usable information

In this section, we first introduce the background of DPO (§2.1). Next, we analyze the discrepancy between the optimization objective of DPO and the actual generation objective, and introduce how the squeezing effect [33] leads to a decrease in the log probability of DPO-preference samples. We then propose a negative gradient constraint method to alleviate the squeezing effect and strengthen the alignment between DPO and the generation objective (§2.2). Finally, we derive the objective of VPO by introducing $\mathcal{V}$-usable information to selectively constrain the negative gradient of non-preference samples from the perspective of informational similarity. (§2.3).

### 2.1    Preliminaries

**Direct preference optimization (DPO)** is a preference optimization method that has gained widespread attention. Instead of learning a reward model, DPO reparameterizes the reward function $r$ in RLHF using a closed-form expression with an optimal strategy, which can be represented as:

$$r(x,y) = \beta \log \frac{\pi_\theta(y|x)}{\pi_{ref}(y|x)} + \beta \log Z(x) \tag{1}$$

where $\pi_\theta$ represents the policy model, $\pi_{ref}$ represents the reference model, $Z(x)$ is the partition function, and $\beta$ is a hyperparameter that controls the deviation between $\pi_\theta$ and $\pi_{ref}$. By combining the reparameterized $r$ with the Bradley-Terry ranking objective [3], $p^*(y_w \succ y_l|x) = \sigma(r^*(x, y_w) - r^*(x, y_l))$, DPO can use the policy model rather than the reward model to represent the probability of preference data. The maximum likelihood objective of the policy model can be represented as:

$$\mathcal{L}_{DPO}(\pi_\theta; \pi_{ref}) = -\mathbb{E}_{(x,y_w,y_l)\sim\mathcal{D}} \left[ \log \sigma \left( \beta \log \frac{\pi_\theta(y_w|x)}{\pi_{ref}(y_w|x)} - \beta \log \frac{\pi_\theta(y_l|x)}{\pi_{ref}(y_l|x)} \right) \right] \tag{2}$$

### 2.2    Negative Gradient Constraint of DPO

As shown in Eq. 2, the objective of DPO is to optimize the margin of the log-likelihood ratio between preference and non-preference samples. However, since the reference model is not involved in the inference process, the reward optimized during DPO training (Eq. 1) does not directly align with the objective of log-likelihood optimization of preference samples during actual inference. Therefore, during DPO training, it is possible for the log probabilities of both preference and non-preference samples to decrease simultaneously (as shown in Figure 1a).

Recently, squeezing effect [33] has been proposed, offering both theoretical and experimental insights into the observed decrease in log probabilities of preference data during DPO training. Specifically, for any model utilizing a softmax output layer to generate probability distributions, when negative

gradients act on low-confidence samples, the following phenomena occur: (1) When model's output distribution $p^t$ exhibits even slight non-uniformity, $p_i$ with smaller $p_i^t$ tend to decrease; (2) When $p^t$ becomes highly concentrated (where the majority of probability mass is captured by $i^*$), then under the influence of negative gradients, all other $p_i(i \neq i^*)$ will decrease, leading to further compression of probability mass toward $i^*$.

Off-policy DPO is more prone to the aforementioned squeezing effect for the following reasons: First, since the preference data is fixed and pre-collected, a distribution shift occurs between the updated policy model and the initial sampling model, resulting in a non-uniform output distribution for preference pairs in the model. Second, due to the discrepancy between the DPO optimization objective and the generation objective, there may be a mismatch between the highest-confidence sample $y^*$ and the preference sample $y_w$. Finally, given that DPO focuses more on avoiding the generation of non-preference responses [11], the non-preference samples $y_l$ are more likely to fall into the low-confidence region of the model's predictions. When a large negative gradient is applied to non-preference samples in the "valley" region, the probability of all responses $y$ except the highest-confidence response $y^*$ is reduced (i.e., the probability mass is pushed towards $y^*$). This may simultaneously decrease the log-probability of both preference and non-preference samples, while increasing the probabilities of certain tokens unrelated to the samples (as shown in Figure 1b).

Compared to incorporating a regularization term into DPO to prevent over-optimization or focusing on increasing the likelihood probability of human preference data, we adopt a negative gradient constraint strategy to mitigate the issue of decreased likelihood probability of preference samples caused by the squeezing effect during the DPO training process. Specifically, we introduce a constraint term $v$ into the DPO loss function:

$$\mathcal{L}_{DPO_{mod}}\left(\pi_\theta; \pi_{ref}\right) = -\mathbb{E}_{(x,y_w,y_l)\sim\mathcal{D}}\left[\log\sigma\left(\beta\log\frac{\pi_\theta\left(y_w|x\right)}{\pi_{ref}\left(y_w|x\right)} - (1-v)\beta\log\frac{\pi_\theta\left(y_l|x\right)}{\pi_{ref}\left(y_l|x\right)}\right)\right]$$
(3)

Where $v$ is a hyperparameter, $1 > v > 0$. To derive the gradients of modified DPO for both preference and non-preference samples, we rewritten the above expression as:

$$L = -\log\sigma(r), \quad r = \beta\log\frac{\pi_\theta(y_w|x)}{\pi_{\text{ref}}(y_w|x)} - (1-v)\beta\log\frac{\pi_\theta(y_l|x)}{\pi_{\text{ref}}(y_l|x)}$$
(4)

By taking the derivatives of the probabilities for the preference and non-preference samples, we obtain the following expressions:

$$\frac{\partial L}{\partial\pi_\theta(y_w|x)} = \frac{\partial L}{\partial r} \cdot \frac{\partial r}{\partial\pi_\theta(y_w|x)} = (\sigma(r) - 1) \cdot \frac{\beta}{\pi_\theta(y_w|x)}$$
(5)

$$\frac{\partial L}{\partial\pi_\theta(y_l|x)} = \frac{\partial L}{\partial r} \cdot \frac{\partial r}{\partial\pi_\theta(y_l|x)} = (1 - \sigma(r)) \cdot \frac{\beta(1-v)}{\pi_\theta(y_l|x)}$$
(6)

When $v > 0$, we can observe that $r$ decreases compared to the original DPO (because the log-likelihood ratio of non-preference samples is negative). This increases the absolute value of $(\sigma(r)-1)$, and thus enhances the gradient of preference samples. Moreover, because the sigmoid function's slope is below 1, the gradient for non-preference samples in Eq.6 monotonically decreases with increasing $v$. This adjustment ensures that $y_w$ is more likely to converge toward the $y^*$ region, while reducing the probability of non-preference samples $y_l$ being trapped in "valley" regions with large gradients, thereby mitigating the squeezing effect.

### 2.3 VPO: Selective Negative Gradient Constraint Based on $\mathcal{V}$-usable information

However, we identify a critical trade-off in the negative gradient constraints of DPO: strong constraints ($v \to 0$) will effectively mitigate the squeezing effect while diminishing the discriminability between preference and non-preference samples; Conversely, weak constraints ($v \to 1$) approximate standard DPO by maintaining discriminative power while failing to alleviate the squeezing effect. This reveals two fundamental limitations: (1) potential performance sub-optimality may be induced by static constraints, and (2) failure to adapt to sample-specific characteristics such as noise or informativeness. These findings highlight the necessity for adaptive constraint mechanisms that can dynamically reconcile the competing objectives of alleviating the squeezing effect and maintaining effective preference learning in DPO.

Since preference and non-preference samples will mutually influence each other during DPO training [33], this interaction inevitably perturbs their intrinsic gradient to some degree. In this section, focusing on reasoning tasks, we propose a selective negative gradient constraint strategy from the perspective of the correlation between preference samples and non-preference samples. We characterize the correlation between texts at two levels: the token-level and the information-level. However, in token-level correlation evaluation, two factors may introduce potential biases: (1) prefix similarity, aligned LLMs tend to generate similar prefixes in reasoning tasks [18, 1]; (2) solution path diversity, reasoning tasks typically admit multiple valid solutions [43]. Therefore, our work focuses on the informational similarity between preference and non-preference samples.

We introduce $\mathcal{V}$-usable information [10, 42] to measure the amount of new label-related information introduced by the reasoning chain of non-preference samples, compared to the input. Specifically, let $X$ and $Y$ represent two random variables, with their sample spaces denoted as $\mathcal{X}$ and $\mathcal{Y}$, respectively. The conditional $\mathcal{V}$-entropy employs a model family $\mathcal{V}$ to learn the mapping from $X$ to $Y$, replacing the conventional conditional entropy which becomes ineffective when the true joint distribution of $X$ and $Y$ is unknown. It is defined as:

$$H_{\mathcal{V}}(Y|X) = \inf_{f \in \mathcal{V}} \mathbb{E}[-\log f[X](Y)] \tag{7}$$

where $f[X]$ yields a probability distribution across the labels. For models $f \in \mathcal{V}$, the goal is to maximize the log-likelihood of label data, both with and without input. Based on the content mentioned above, $\mathcal{V}$-usable information [42] is proposed, which generalizes the Shannon information [35] to quantify the information about $Y$ that can be extracted from $X$ under model family $\mathcal{V}$, denoted as $I_{\mathcal{V}}(X \to Y)$, and is defined as follows:

$$I_{\mathcal{V}}(X \to Y) = H_{\mathcal{V}}(Y \mid \varnothing) - H_{\mathcal{V}}(Y \mid X) \tag{8}$$

Furthermore, Pointwise $\mathcal{V}$-Information (PVI) [10] is proposed, which extends the $\mathcal{V}$-usable information framework from the dataset level to the instance level. [24] further extends this to evaluation in different contexts, which is defined as:

$$\text{PVI}(x \to y) = -\log g[\varnothing](y) + \log g[x](y) \tag{9}$$

Following previous studies [30, 39], We use PVI to quantify the amount of label-related information introduced by the reasoning chain of non-preference samples under the policy model $\pi_0$ in the initial state, beyond the input X. We define it as $\text{PVI}_l$, which can be expressed as:

$$\text{PVI}_l = \text{PVI}(c_l \to y|x) = -\log \pi_0\,(y|x) + \log \pi_0\,(y|x, c_l) \tag{10}$$

where $c_l$ is the reasoning chain in non-preference samples. For the preference sample, it can be expressed as: $\text{PVI}_w = \text{PVI}(c_w \to y|x)$. Notably, nearly all preference samples' reasoning chains contain positive label-related information ($\text{PVI}_w > 0$). We categorize non-preference samples based on their $\text{PVI}_l$ into the following two cases: (1) If $\text{PVI}_l > 0$, it indicates a stronger correlation with preference samples (since all their PVI are positive, reflecting that both can introduce additional positive label-relevant information) and a stronger interaction between them. At this point, the non-preference sample is somewhat 'pulled up' by the positive gradient (weaker than the positive gradient of the preference samples), which constrains its own negative gradient and alleviates the squeezing effect. In this case, we do not apply constraints to the negative gradient to ensure the distinction between preference and non-preference samples (i.e., $v = 0$). (2) If $\text{PVI}_l < 0$, it indicates a weaker correlation with the information of the preference samples and a reduced interaction between them. The non-preference sample still maintains a large negative gradient, making the squeezing effect more likely. In this case, we constrain the negative gradient based on the normalized $\text{PVI}_l$. As the $\text{PVI}_l$ decreases (correlation weakens), the gradient constraint on the negative sample increases (i.e., $v = -\text{PVI}_l$). Based on this, we can derive the objective of VPO as:

$$\mathcal{L}_{VPO}\,(\pi_{\theta}; \pi_{ref}) = -\mathbb{E}_{(x, y_w, y_l) \sim \mathcal{D}} \left[ \log \sigma \left( \beta \log \frac{\pi_{\theta}(y_w|x, c_w)}{\pi_{ref}(y_w|x, c_w)} - \beta(1 - v) \log \frac{\pi_{\theta}(y_l|x, c_l)}{\pi_{ref}(y_l|x, c_l)} \right) \right] \tag{11}$$

where $v$ is calculated as :

$$v = \begin{cases} 0, & \text{PVI}_l > 0 \\ \sigma(-\text{PVI}_l), & \text{PVI}_l < 0 \end{cases} \tag{12}$$

In summary, VPO employs $\mathcal{V}$-usable information to measure the information similarity between preference and non-preference samples, selectively constraining the negative gradient of non-preference samples. This helps to better coordinate and balance the competing objectives of alleviating the squeezing effect and preserving effective preference learning in DPO.

## 3  Experimental Setup

**Models and training settings.** We use two types of models: Llama-3.1-8B [14] and Qwen-2.5-7B [31], and perform preference optimization under two model settings: base model and instruction-tuned model. VPO focuses on optimizing reasoning tasks, so we use tasks that require the model to perform intermediate reasoning (i.e., generating a reasoning chain before answering, otherwise performance would be poor), including: (1) GSM8K [8], a dataset containing real, high-quality elementary school math application problems; and (2) MATH [17], a dataset containing challenging mathematics competition problems. Each question in both datasets contains a problem $x$ and the final numerical answer $y$, and both datasets have 7.5k questions in their training sets.

We use the few-shot prompt for sampling, which includes the problem, reasoning chain, and answer, and follows a specific format to allow the extraction of the predicted answer later (the exact prompt can be found in Appendix A.1). For each model used, we sample $N = 10$ solutions for each question, with a temperature setting of 0.8. Following previous studies [29], we classify responses that match the predicted answer with the label as preference samples, and those that do not match as non-preference samples, which can be expressed as:

$$D_i^w = \{c_i^n, y_i^n, x_i^n \mid r_i^n = 1\} \quad D_i^l = \{c_i^n, y_i^n, x_i^n \mid r_i^n = 0\}$$

When all 10 solutions sampled by the LLM are either entirely correct or entirely incorrect, we discard the query. We then simultaneously select samples from $D_i^w$ and $D_i^l$ to generate $K$ pairs of indices, thereby constructing the preference pair dataset:

$$D^{pairs} = \left\{ (c_i^{w_k}, y_i^{w_k}), (c_i^{l_k}, y_i^{l_k}) \,\Big|\, \forall x_i \in D \text{and } k \in [K] \right\}$$

For Llama 3.1-8B-Base, Llama-3.1-8B Instruct, and Qwen-2.5-7B-Base, we retain 5 preference pairs for each query. For Qwen-2.5-7B-Instruct, we retain 10 preference pairs for each query. In total, the training data constructed for each model contains 30k-40k sample pairs.

**Evaluation benchmarks.** We evaluate the model's performance on standard mathematical reasoning benchmarks, including: (1) GSM8K [8]; (2) MATH500 [17]; (3) OlympiadBench-Math [16]; (4) AMC23; (5) Minerva Math [20].

**Baselines.** We compare VPO with several offline preference optimization methods. RPO [29, 23] introduces a negative log-likelihood term to regularize the policy model, aiming to improve performance on inference tasks. SimPO [25] uses average log-probability of a sequence as an implicit reward, making it more aligned with the generation objectives and eliminating the need for a reference model. IPO [2] avoids the assumptions of DPO by replacing pairwise preferences with pointwise rewards. It regularizes the policy model towards the reference model by controlling the gap between the log-likelihood ratios. TDPO [45] is a token-level optimization strategy that uses the Bradley-Terry model to build a token-based reward system, enhancing the control over the KL divergence. We list the optimization objectives and training details of different baselines in Appendix A.2.

## 4  Experimental Results

In this section, we present the main results of our experiments, highlighting the superior performance of VPO across different settings. Subsequently, we conduct ablation experiments on VPO, demonstrating the effectiveness of the negative gradient constraint method and the selective constraint method using PVI (§4.1). Finally, we provide an in-depth understanding and analysis of VPO (§4.2).

### 4.1  Main results and Ablation study

**VPO consistently outperforms existing preference optimization methods across different settings.** As shown in Table 1, VPO significantly improves performance over DPO across all benchmarks and settings by selectively constraining the negative gradients of non-preference samples from the perspective of information similarity. Moreover, we can observe that VPO achieves the best overall performance across all benchmark tests and settings. Specifically, VPO outperforms all variants of DPO on Qwen-2.5-7B-Base. Compared to the best-performing baseline, VPO achieves a 2.4% improvement on MATH500, a 3.26% improvement on Olympiad MATH, and a 12.05% improvement on AMC23. The same trend is also observed in the experimental results for other models. For

Table 1: Results of VPO, DPO and its variants on diverse mathematical reasoning tasks. The best results are highlighted in **bold**, while the second-best ones are underlined.

| Method | Qwen2.5-7B-Base | | | | | | Qwen2.5-7B-Instruct | | | | | |
| --- | --- | --- | --- | --- | --- | --- | --- | --- | --- | --- | --- | --- |
| | MATH 500 | GSM8k | Minerva MATH | Olympiad MATH | AMC 23 | Avg | MATH 500 | GSM8k | Minerva MATH | Olympiad MATH | AMC 23 | Avg |
| Base | 59.00 | 79.98 | 15.07 | 21.93 | 18.07 | 38.81 | **73.20** | 84.23 | 27.94 | 36.44 | 44.58 | 53.28 |
| DPO | 61.00 | 80.89 | 21.32 | 27.11 | 32.53 | 44.57 | 45.60 | 75.66 | **28.31** | 33.63 | 44.58 | 45.56 |
| TDPO | 59.20 | 79.68 | 17.28 | 26.22 | 28.92 | 42.26 | 48.00 | 77.33 | 23.53 | 20.15 | 34.94 | 40.79 |
| SimPO | 64.60 | 74.15 | 20.59 | 26.07 | 33.73 | 43.83 | 43.80 | 72.86 | 19.85 | 14.52 | 18.07 | 33.82 |
| IPO | 51.80 | 75.51 | 15.44 | 23.41 | 32.53 | 39.74 | 71.20 | 84.99 | 26.84 | **37.19** | 44.58 | 52.96 |
| RPO | 66.40 | 84.46 | 21.69 | 27.26 | 31.33 | 46.23 | 56.20 | 81.27 | 27.81 | 33.93 | 39.76 | 47.79 |
| VPO | **68.80** | **84.91** | **23.89** | **30.52** | **45.78** | **50.78** | 71.60 | **86.73** | **28.31** | 36.44 | **48.19** | **54.26** |

| Method | Llama-3.1-8B-Base | | | | | | Llama-3.1-8B-Instruct | | | | | |
| --- | --- | --- | --- | --- | --- | --- | --- | --- | --- | --- | --- | --- |
| | MATH 500 | GSM8k | Minerva MATH | Olympiad MATH | AMC 23 | Avg | MATH 500 | GSM8k | Minerva MATH | Olympiad MATH | AMC 23 | Avg |
| Base | 17.40 | 55.80 | 0.37 | 0.15 | 0.00 | 14.74 | 45.00 | 80.52 | **22.43** | 15.26 | 27.71 | 38.18 |
| DPO | 10.00 | 54.51 | 4.04 | 1.93 | 2.41 | 14.58 | 18.40 | 54.51 | 9.93 | 5.48 | 7.23 | 19.11 |
| TDPO | 14.80 | 59.29 | 1.47 | 1.33 | 0.00 | 15.38 | 22.75 | 73.09 | 12.50 | 6.52 | 6.02 | 24.18 |
| SimPO | 19.20 | 55.88 | **8.46** | 1.63 | 4.82 | 18.00 | 31.80 | 74.60 | 10.66 | 7.70 | 15.66 | 28.09 |
| IPO | 3.80 | 61.94 | 0.00 | 0.15 | 1.20 | 13.42 | **47.20** | 81.35 | 20.22 | **15.41** | 25.30 | 37.90 |
| RPO | 19.60 | **65.14** | 7.35 | 2.37 | 8.43 | 20.58 | 31.20 | 81.80 | 14.71 | 9.48 | 7.23 | 28.88 |
| VPO | **20.80** | 63.84 | 6.62 | **3.56** | 8.43 | **20.65** | 46.40 | **83.62** | 20.96 | **15.41** | **30.12** | **39.30** |

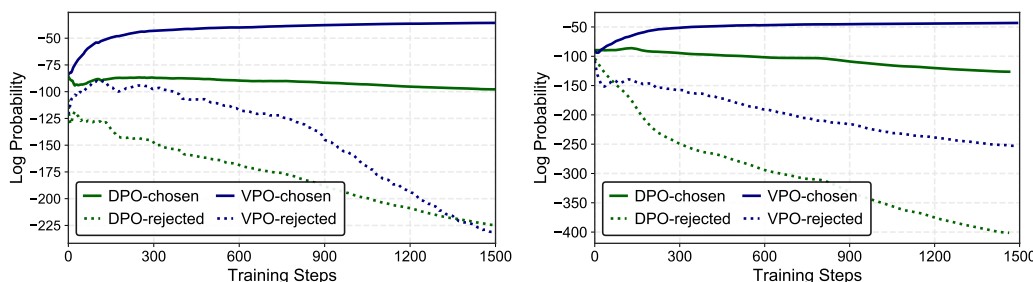

Figure 2: The log probability change curves of preference (chosen) and non-preference (rejected) samples for VPO and DPO across different models. Left: Llama-3.1-8B-Base, Right: Llama-3.1-8B-Instruct.

example, on AMC23, Qwen2.5-7B-Instruct and Llama-3.1-8B-Instruct show improvements of 3.61% and 2.41%, respectively, compared to the best-performing baseline.

**VPO effectively alleviates the squeezing effect.** From Table 1, we observe a decline in the model's performance after DPO training on the Instruct model. We present the change curves of the log probabilities for preference and non-preference samples under DPO and VPO on Llama-3.1-8B-Base and Llama-3.1-8B-Instruct in Figure 2. As shown in Figure 2, due to the squeezing effect, DPO exhibits a decrease in the log probability of preference samples during training, deviating from the objective of optimizing the log probability of preference samples during inference. This issue becomes more pronounced on Llama-3.1-8B-Instruct. Notably, VPO improves the log probability of preference samples on both the Base and Instruct models. For Base models, the margin between preference and non-preference samples is even larger than in DPO.

For Instruct models, while the constraint on negative gradients slows the decline of log-probability for non-preference samples, we observe no significant difference in log probability gaps between preference pairs in VPO (logp gap = 244) and DPO (logp gap = 274). Qwen-2.5 series model exhibits the same trend, as detailed in Appendix B.1. This indicates that VPO can effectively mitigate the squeezing effect without compromising DPO's core optimization objective.

**Both key designs in VPO are crucial.** In the Table 2, we perform an ablation analysis of the key components of VPO: (1) negative gradient constraint; (2) sample information measurement (i.e., the calculation of $PVI_l$). We set $v$ to range from 0.1 to 0.9 to compare the DPO under the negative gradient constraint with the original DPO. To ensure the distinctiveness of the experimental results,

Table 2: Performance comparison of DPO vs VPO across diverse math benchmarks under varying $v$-constraints. The best results are highlighted in **bold**, while the second-best ones are underlined.

| Method | Llama-3.1-8B-Instruct | | | | | | Qwen2.5-7B-Base | | | | | |
|---|---|---|---|---|---|---|---|---|---|---|---|---|
| | MATH 500 | GSM8k | Minerva MATH | Olympiad MATH | AMC 23 | Avg | MATH 500 | GSM8k | Minerva MATH | Olympiad MATH | AMC 23 | Avg |
| Base | 45.00 | 80.52 | **22.43** | 15.26 | 27.71 | 38.18 | 59.00 | 79.98 | 15.07 | 21.93 | 18.07 | 38.81 |
| DPO | 18.40 | 54.51 | 9.93 | 5.48 | 7.23 | 19.11 | 61.00 | 80.89 | 21.32 | 27.11 | 32.53 | 47.58 |
| 0.1 | 22.80 | 64.06 | 13.97 | 6.37 | 6.02 | 22.64 | 67.60 | 84.46 | 22.79 | 29.19 | 31.33 | **51.01** |
| 0.2 | 21.80 | 67.55 | 11.76 | 6.81 | 12.05 | 24.00 | 68.60 | 84.84 | 20.59 | 29.04 | 39.76 | 50.77 |
| 0.3 | 25.00 | 70.74 | 15.44 | 6.96 | 9.64 | 25.56 | **68.80** | 84.15 | 20.96 | 29.48 | 38.55 | 50.85 |
| 0.4 | 31.60 | 76.50 | 15.81 | 9.33 | 10.84 | 28.82 | 68.60 | 83.40 | 20.59 | 29.33 | 43.37 | 50.48 |
| 0.5 | 34.20 | 74.75 | 16.91 | 11.85 | 12.05 | 29.95 | 68.40 | 83.62 | 21.32 | 28.59 | 39.76 | 50.48 |
| 0.6 | 39.20 | 76.50 | 17.65 | 12.59 | 18.01 | 32.79 | 66.60 | 85.67 | 20.22 | 27.20 | 40.96 | 49.92 |
| 0.7 | 44.80 | 79.53 | 18.01 | 13.78 | 14.46 | 34.12 | 65.60 | **86.28** | 20.59 | 28.89 | 37.35 | 50.34 |
| 0.8 | 44.40 | 77.18 | 19.12 | 14.07 | 25.30 | 36.01 | 65.80 | 85.97 | 19.49 | 27.56 | 42.17 | 49.70 |
| 0.9 | 16.40 | 48.90 | 0.74 | 4.41 | 1.20 | 14.33 | 66.80 | 85.37 | 20.22 | 27.56 | 39.76 | 49.99 |
| VPO | **46.40** | **83.62** | 20.96 | **15.41** | **30.12** | **39.30** | **68.80** | 84.91 | **23.89** | **30.52** | **45.78** | **52.03** |

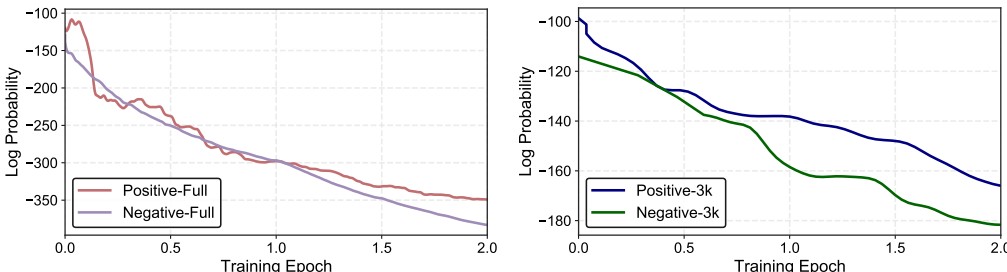

Figure 3: Decline curves of log-probabilities for non-preference samples under different configurations. Left: positive and negative non-preference samples in full training. Right: independent training on 3k preference pairs consisting exclusively of positive and negative non-preference samples.

we choose Qwen-2.5-7B-Base to replace Llama-3.1-8B-Base as the base model. Experimental results show that negative gradient constraints at almost all levels lead to performance improvements compared to the original DPO. For example, for Llama-3.1-8B-Instruct, when $v = 0.3$, compared to DPO, there are improvements of 26.4%, 25.20%, 8.08% on MATH500, GSM8k, Minerva MATH, respectively. We present the trend of changes in preference and non-preference samples with different negative gradient constraint intensities in the Appendix B.3. Besides, we can observe that the optimal constraint intensity varies for different models. For example, for Llama-3.1-8B-Instruct, the best performance is achieved when $v \in [0.7, 0.8]$, while for Qwen2.5-7B-Base, the optimal performance occurs when $v \in [0.1, 0.3]$. Notably, VPO consistently outperforms DPO even at its optimal negative gradient constraint levels across different models. This validates the effectiveness of employing $\text{PVI}_l$ for selective negative gradient constraint on non-preference samples.

## 4.2 In-depth analysis of VPO

We define non-preference samples into two categories: samples with a high correlation to preference samples (positive, $\text{PVI}_l > 0$) and samples with a low correlation to preference samples (negative, $\text{PVI}_l < 0$).

**$\text{PVI}_l$ can effectively measure the correlation between preference and non-preference samples.** We design the following experiment:(1) track the log probabilities of positive and negative non-preference samples during the full data training. (2) to avoid mutual interference between positive and negative non-preference samples during the full data training, we separately train two subsets: 3k preference pairs containing either positive-only or negative-only non-preference samples, selected from the preference dataset. The experimental results presented in Figure 3 demonstrate that, both in full dataset training and in separate training, the decline rate of the positive non-preference samples is slower than that of the negative non-preference samples. This trend strongly supports our analysis in section §2.3, namely that due to the positive gradient influence of preference samples, the negative

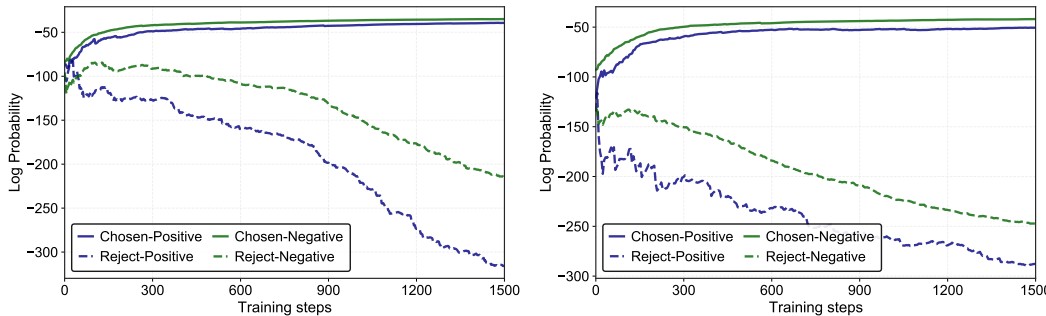

Figure 4: The log probability change curves of preference pairs under negative gradient constraint (negative) and unconstrained (positive) conditions during VPO training. Left: Llama-3.1-8B-Base, Right: Llama-3.1-8B-Instruct.

Table 3: Results of VPO, DPO, and their variants on Qwen3-14B-Base across various mathematical reasoning tasks. The dashed line represents the Negative Gradient Constraint method based on different similarity metrics: Embedding Cosine similarity and Jaccard-based textual similarity.

| Method | MATH500 | GSM8k | Minerva MATH | Olympiad MATH | AMC23 | AIME24 | Avg |
|--------|---------|-------|--------------|---------------|-------|--------|-----|
| Base | 63.60 | 93.93 | 24.63 | 21.78 | 22.89 | 0.00 | 37.81 |
| DPO | 76.40 | 94.09 | 28.68 | 33.63 | 45.78 | 20.00 | 49.76 |
| TDPO | 70.40 | 94.31 | 26.47 | 27.56 | 38.55 | 13.33 | 45.10 |
| Simpo | 75.60 | 95.75 | 31.25 | 32.44 | 43.37 | 16.67 | 49.18 |
| IPO | 64.60 | 94.24 | 25.00 | 22.81 | 22.89 | 10.00 | 39.92 |
| RPO | 78.00 | 95.68 | 32.35 | 34.67 | 51.81 | 13.33 | 50.97 |
| *Negative Gradient Constraint* | | | | | | | |
| Cosine | 75.60 | 95.75 | 29.78 | 34.67 | 44.58 | 20.00 | 50.06 |
| Jaccard | 78.00 | 95.98 | 32.35 | **37.78** | 49.40 | 16.67 | 51.70 |
| **VPO** | **79.00** | **96.06** | **35.66** | 35.41 | **53.01** | **26.67** | **54.30** |

gradient of positive non-preference samples is constrained, causing their log-probability to decrease more slowly than that of negative non-preference samples. This demonstrates the effectiveness of $PVI_l$ in quantifying the information correlation between preference and non-preference samples.

**Trend of log-probability changes for different preference pairs in VPO training.** Figure 4 shows the log-probability trajectories during VPO training for preference pairs containing positive non-preference samples ($v = 0$) and preference pairs containing negative non-preference samples ($v > 0$) under the full dataset. We present the results of the Qwen2.5 model in Appendix B.2. We can observe that, under varying configurations, negative non-preference samples exhibit slower log probability decline, while the log probability of their corresponding preference samples is increased. Notably, even in the absence of negative gradient constraints on positive non-preference samples, their corresponding preference samples still exhibit log-probability enhancement. We attribute this improvement to two key factors: (1) the intrinsic positive gradients from samples themselves combined with the "pull-up" effect from reinforced positive gradients of similar preference samples, and (2) the strong correlation between positive non-preference samples and preference samples, which constrains the negative gradient of non-preference samples, prevents their log probability from decreasing sharply (Fig.3), thereby restrains the squeezing effect (Fig.1c). We can also observe that the log-probability gap between positive non-preference samples and their preference samples remains significant. These results suggest that the selective negative gradient constraint on non-preference samples in VPO effectively restrains the sequeezing effect, better aligning with the generation objective, while preserving the distinguishability between strongly correlated sample pairs.

### 4.3 Ablation Study

**The performance of VPO on different model size.** To evaluate the impact of model size on different preference optimization algorithms, we also conducted experiments on Qwen3-14B-Base. The experimental results are shown in Table 3. The results indicate that, on the larger-scale Qwen3-

14B model, VPO outperforms other preference optimization algorithms across all datasets. This demonstrates the performance stability of VPO across different model scales.

**The effect of Negative Gradient Constraint combined with different correlation evaluation methods.** To further investigate the effect of Negative Gradient Constraint combined with different correlation evaluation methods, we compare VPO (based on PVI) with methods based on Embedding Cosine similarity and Jaccard-based textual similarity. The experimental results in Table 3 show that the PVI-based Negative Gradient Selective Constraint method outperforms other similarity evaluation methods. We attribute this to the fact that, compared to other embedding and text similarity-based metrics, V-available information can capture deeper language structures and semantic relationships, providing interpretable insights into causal relationships and dependency directions. With these advantages, VPO achieves a more significant performance improvement.

**The effectiveness of VPO in complex common-sense reasoning.** To explore reasoning abilities beyond mathematics, we use the ARC dataset [7], which covers multiple scientific domains. The dataset contains 7.7k questions, divided into an easy set and a challenge set. We sample 3.37k training samples from both the easy and challenge sets, performing 30 samplings per query to construct a preference dataset of approximately 10k pairs. We test on the ARC-Challenge test set, which contains 1,172 questions. Figure 5 shows that VPO continues to demonstrate superior performance on this dataset, proving its effectiveness beyond mathematical reasoning tasks.

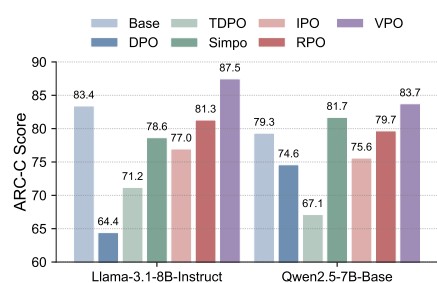

Figure 5: Performance comparison of different methods on ARC-Challenge.

## 5  Limitations

By applying selective negative gradient constraints to different non-preference samples, VPO can mitigate the squeezing effect caused by large negative gradients, enhance the log probability of preference samples, and prevent the issue of a small log probability gap between preference samples due to excessive constraints on negative gradient. Through various experiments, we demonstrate the rationale and effectiveness of the proposed VPO. However, our method is not without its flaws. In detail, since $PVI_l$ is typically not between 0 and 1, we simply use a sigmoid function for normalization. Future research could explore other normalization functions and investigate the effect of normalizing the loss function after considering the interaction between $PVI_l$ and the PVI of preference samples.

## 6  Conclusion

In this work, we propose VPO, an efficient reasoning preference optimization method. By using $\mathcal{V}$-usable information to selectively constrain the negative gradient of non-preference samples from the perspective of information similarity, VPO can alleviate the issue of log probability reduction for preference samples, enhance alignment with the generation target, and maintain the model's ability to distinguish between preference and non-preference samples. Compared to existing methods, VPO consistently achieves better overall performance across various training settings. Extensive analysis shows that the negative gradient constraint and the information similarity measure design in VPO are crucial, validating the rationale and effectiveness of VPO.

**Acknowledgments and Disclosure of Funding**

This work is supported by the National Key RD Program of China (Grant No. 2023YFB3307500). This work is also supported the National Natural Science Foundation of China (Grant No. 62306087), the Natural Science Foundation of Shandong Province (Grant No. ZR2023QF154), National Natural Science Foundation of Shandong Province (No.ZR2025MS997), Special Funding Program of Shandong Taishan Scholars Project, CCF-Sangfor 'Yuanwang' Research Fund (Grant No. 20240201) and CCF-Tencent Rhino-Bird Open Research Fund (CCF-Tencent RAGR20250105).

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

# A  Details on Experimental Setup

## A.1  Prompts

**GSM8K.** For each GSM8K question, we use the following prompt as the input to the LLM. Specifically, the prompts include four fixed context examples selected from [12].

```
Your task is to answer the question below. Give step by step reasoning
before you answer and when you're ready to answer, use the format "The
answer is ..."

Question: [question for the first example]
Let's think step by step [solution for the first example]
The answer is [answer (e.g., number) here]

Question: [question for the second example]
Let's think step by step [solution for the second example]
The answer is [answer (e.g., number) here]

Question: [question for the third example]
Let's think step by step [solution for the third example]
The answer is [answer (e.g., number) here]

Question: [question for the fourth example]
Let's think step by step [solution for the fourth example]
The answer is [answer (e.g., number) here]

Question: [the question to be solved]
```

**MATH.** For each MATH question, we use the following prompt as the input to the LLM. Specifically, the prompts include four fixed context examples selected from [12], and the format of demonstrations in Few-shot is consistent with that of GSM8K.

```
Your task is to answer the last question below. Give step by step reasoning
before you answer, and when you're ready to answer, please wrap your answer
in \\boxed, and conclude using the format "The answer is ..."
```

**ARC.** For each ARC question, we use the following prompt as input for the LLM, assuming the question has four options.

```
Your task is to answer the question below. Give step by step reasoning
before you answer, and when you're ready to answer, conclude using the
format "Final answer: (insert letter here)"
Question: [question here]
(A) [option A here]
(B) [option B here]
(C) [option C here]
(D) [option D here]
```

## A.2  Optimization Objectives of Different Baselines and Training Details

The optimization objectives of different off-policy preference optimization methods compared in the experiment are shown in Table 4. For training, we perform full-parameter preference optimization training on the model, training all baseline methods for 2 epochs with a learning rate of $5 \times 10^{-7}$. The coefficient $\beta$ in the DPO loss is tuned in $\{0.05, 0.1, 0.5, 1.0\}$, and we end up using 0.05 in this experiment. For the parameters $\gamma$ and $\gamma/\beta$ in SimPO, we tried the following combinations: $\{2.0, 0.5\}$, $\{2.5, 0.55\}$, $\{10, 0.3\}$, $\{10, 0.5\}$, and $\{10, 0.1\}$. Ultimately, we select $\{10, 0.5\}$ to train all models.

## A.3  Computation Environment.

All experimental results in this paper were conducted on GPUs with 8×H800 and 8×H20 configurations, with the same GPU model used across all experiments in each set.

Table 4: Optimization Objectives of Different Baselines

| Method | Objective |
|---|---|
| DPO | $-\log \sigma \left( \beta \log \frac{\pi_\theta(y_w|x)}{\pi_{\text{ref}}(y_w|x)} - \beta \log \frac{\pi_\theta(y_l|x)}{\pi_{\text{ref}}(y_l|x)} \right)$ |
| TDPO | $\log \sigma \left( \beta \left( \log \frac{\pi_\theta(y_w|x)}{\pi_{\text{ref}}(y_w|x)} - \log \frac{\pi_\theta(y_l|x)}{\pi_{\text{ref}}(y_l|x)} - \left( D_{\text{SeqKL}}(x, y_l; \pi_{\text{ref}}\|\pi_\theta) - D_{\text{SeqKL}}(x, y_w; \pi_{\text{ref}}\|\pi_\theta) \right) \right) \right)$ |
| SimPO | $-\log \sigma \left( \frac{\beta}{|y_w|} \log \pi_\theta(y_w|x) - \frac{\beta}{|y_l|} \log \pi_\theta(y_l|x) - \gamma \right)$ |
| IPO | $\left( \log \frac{\pi_\theta(y_w|x)}{\pi_{\text{ref}}(y_w|x)} - \log \frac{\pi_\theta(y_l|x)}{\pi_{\text{ref}}(y_l|x)} - \frac{1}{2\tau} \right)^2$ |
| RPO | $-\log \sigma \left( \beta \log \frac{\pi_\theta(y_w|x)}{\pi_{\text{ref}}(y_w|x)} - \beta \log \frac{\pi_\theta(y_l|x)}{\pi_{\text{ref}}(y_l|x)} \right) - \log \frac{\pi_\theta(y_w|x)}{|y_w|}$ |
| VPO | $\log \sigma \left( \beta \log \frac{\pi_\theta(y_w|x)}{\pi_{ref}(y_w|x)} - \beta(1-v) \log \frac{\pi_\theta(y_l|x)}{\pi_{ref}(y_l|x)} \right)$ |

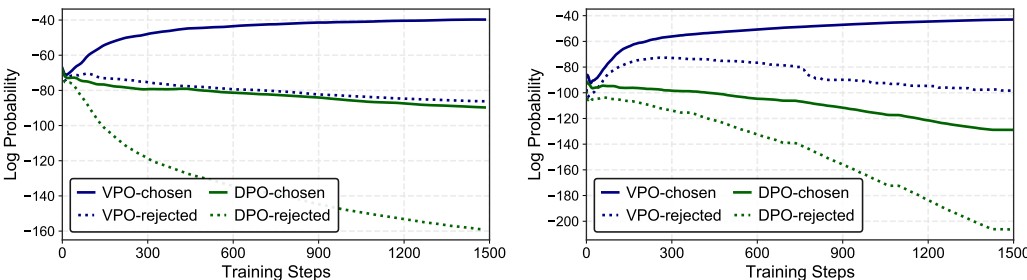

Figure 6: The log probability change curves of preference (chosen) and non-preference (rejected) samples for VPO and DPO across different models. Left: Qwen2.5-7B-Base, Right: Qwen2.5-7B-Instruct.

## B  Log Probability Change Curves of VPO and DPO Preference Pairs under Different Experimental Settings

### B.1  Log Probability Change Curves of VPO and DPO Preference Pairs on the Qwen2.5 Model

In Figure 6, we present the log probability change curves of preference and non-preference samples for DPO and VPO on the Qwen-2.5-7B-Base model and the Qwen-2.5-7B-Instruct model. We can observe that, similar to the Llama 3.1 model, VPO is still able to enhance the log probability of preference samples on the Qwen 2.5 model. From Figure 2, we can see that compared to the Llama-3.1 model, the Qwen-2.5 model exhibits a relatively smaller log probability difference between preference and non-preference samples during training. We can observe that, although the log probability of non-preference samples in VPO decreases only slightly during training, there remains a certain difference between preference and non-preference samples. Specifically, the maximum log probability difference of the preference pairs in Qwen 2.5-7B-Instruct is 56 for VPO and 72 for DPO; for the Qwen 2.5-7B-Instruct preference pairs, the maximum log probability difference is 48 for VPO and 69 for DPO.

### B.2  Log Probability Change Curves of different preference pairs for VPO on the Qwen2.5 model.

As shown in Figure 7, on both the Qwen2.5-7B-Base model and the Qwen2.5-7B-Instruct model, VPO can slow down the decrease in the log-probability of negative non-preference samples while boosting the log-probability of their corresponding preference samples. For unconstrained positive non-preference samples, VPO can still enhance the log-probability of preference samples, as analyzed in section 4.2, while maintaining the log-probability difference between positive non-preference samples and preference samples. These results indicate that the selective negative gradient constraint

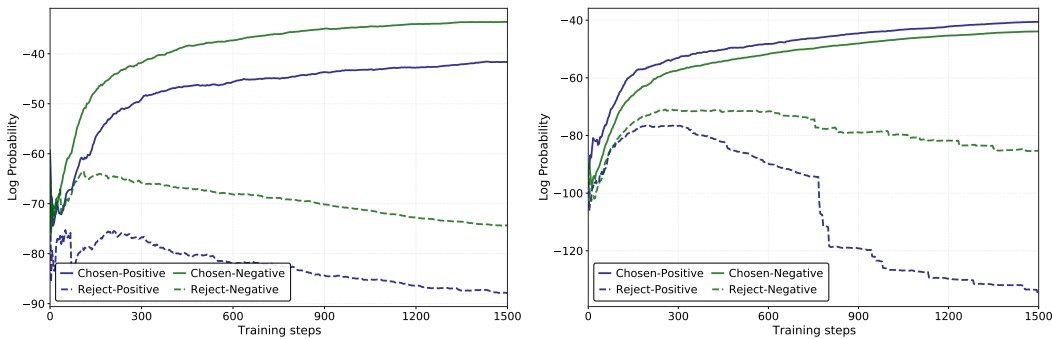

Figure 7: The log probability change curves of preference pairs under negative gradient constraint (negative) and unconstrained (positive) conditions during VPO training. Left: Qwen2.5-7B-Base, Right: Qwen2.5-7B-Instruct.

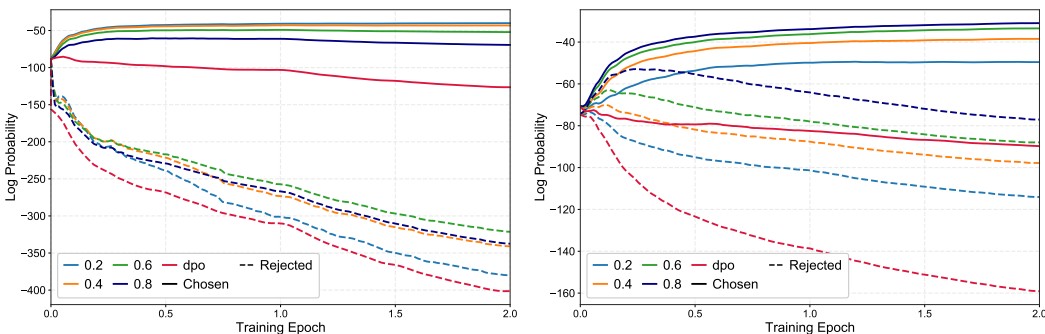

Figure 8: The log probability change curves of preference and non-preference samples under different strengths of negative gradient constraints in DPO. Left: Llama-3.1-8B-Instruct, Right: Qwen2.5-7B-Base.

on non-preference samples in VPO effectively suppresses the squeezing effect, allowing it to better align with the generation target while preserving the distinguishability between strongly correlated preference pairs.

### B.3 Comparison of Log Probability Curve Changes of DPO under Different Negative Gradient Constraint Intensities.

Figure 8 shows the log probability change curves of preference and non-preference samples under different levels of negative gradient constraints for DPO. To clearly illustrate the changes in the curves, we only present the log probability change curves for five cases: $v = \{0.2, 0.4, 0.6, 0.8\}$ and DPO ($v = 0.0$). As shown in Figure 8, the log probabilities of preference samples exhibit a monotonic increase during DPO optimization with negative gradient constraints ($v > 0$). However, it is noteworthy that when $v \to 1$ the probability discriminability between preference pairs significantly diminishes due to intensified constraints. Conversely, as $v \to 0$, the model reduces to standard DPO, where the log probability of preference samples decreases, and the squeezing effect is more likely to occur.

## C   Related Work

**Offline preference optimization.** Aligning LLMs with human preferences and values is a key component of post-training for LLMs. RLHF is a technique used to ensure that LLMs align with human preferences [27, 36, 6]. RLHF typically consists of three stages: supervised fine-tuning [46], reward model training [13, 21], and policy optimization [34]. This algorithm is complex and challenging to optimize. DPO reparameterizes the reward function of RLHF to obtain a closed-form expression for the optimal policy, enabling it to directly learn the policy model from preference data

without the need for an explicit reward model. However, DPO lacks the ability to sample preference pairs from the optimal policy model, preventing the policy model from receiving immediate feedback on its generated content and causing a distribution shift between the initial and aligned policy models, turning the alignment process into off-policy learning. To address this issue, existing methods extend to iterative training frameworks, where the reference model is continuously updated with the latest policy model to generate new preference pairs in each iteration, or LLMs are used as annotators to provide online feedback during training iterations [29, 15, 41, 40]. In this study, we focus solely on the offline scenario of DPO, avoiding any iterative training processes.

**Reasoning preference optimization.** DPO has been successful in general instruction tasks, but it still faces challenges when applied to reasoning and mathematical problems [11, 5, 19]. While several methods have been proposed to curate or distill training data for the reasoning task [37, 44], this study focuses more on the optimization of the algorithm itself. Several research methods aim to achieve preference optimization while maintaining a high generation probability for preference examples. For instance, RPO [29, 23] add the negative log-likelihood of preference samples to DPO, explicitly regularizing the policy model to mimic the initial policy model. [22] uses contrastive estimation to identify key tokens in the reasoning chain that significantly impact incorrect results, providing token-level signals for preference optimization. IPO [2] regularizes the policy model to the reference model by controlling the gap between the log-likelihood ratios. In contrast, our research focuses on selectively constraining the negative gradient of DPO to mitigate the squeezing effect, adjust the preference optimization direction, and thereby improve the alignment between the DPO optimization objective and the actual generation target.

