# OpenReview forum: "VPO: Reasoning Preferences Optimization Based on $\mathcal{V}$-Usable Information"
_NeurIPS.cc/2025/Conference — NeurIPS 2025 spotlight_

### Official Review · Reviewer_1j6X · 2025-07-02

**Clarity:** 4
**Significance:** 4
**Originality:** 4
**Rating:** 6
**Confidence:** 3

**Summary:**

The paper sets the stage by identifying an issue in the DPO training process, where when you optimize on low confidence preference pairs, there exists a squeezing effect identified in previous papers, which reduces the log probabilities of both positive and negative samples. It also skews the output probability distribution drastically which is not helpful or easy to recover from. The paper tries to explain the underlying reasons which contribute to such a problem and attempts to constrain the gradients in such a case so as to minimize this effect. The authors also provide strong derivations showing the effect their proposed methodology provides.

In detail, when using a softmax function the output and if the confidence lies in the valley region of the function, a negative gradient will suppress the entire distribution, including both positive and negative samples. This is especially made worse because the data is collected off-policy earlier own, hence the updates made may not fully be considered on-policy and hence diverges. There are also forces which makes it likely to fall into the valley region also.

Towards fixing this, the authors introduce a regularization term in the DPO loss, with an added hyperparameter v. The authors derived the effect this has on the gradients. This counteracts the above issue so that it avoids the problem. The regularization term enhances the gradients of the preference samples whereas reducing the probability of non-preference samples. This avoids the squeezing effect.

Also the authors go into deciding how to set the hyperparameter appropriately by using the PVI information of the preference pairs. Two cases are considered depending on PVI of non-preference samples falling into two regimes, positive and negative, which are considered separately to set the hyperparameter appropriately.

The experiments are run on models Qwen2.5-7B and Llama-3.1-8B Base & Instruct on a set of relevant benchmarks, also compared with appropriate baselines also. Overall the results seem to suggest quite a good lift compared to other baselines. The paper also goes into in depth analysis of VPO.

**Questions:**

Generally I feel the paper is quite thorough. I had clarifications on the following subtle statements which are made, which were not quite obvious to me.

- “The issue of preference sample probability decline in DPO becomes more severe in reasoning tasks”. Could you shed some light on this claim? It was not obvious to me on from the referred papers.
- "Besides, DPO focuses more on how to avoid generating non-preference samples , which makes non-preference samples more likely to fall into the “valley” region under the influence of the negative gradient". Could you explain this statement? How does avoiding generating non-preferences samples, make it more likely to fall into the valley region.

The above questions are answered satisfactorily by the authors.

**Ethical Concerns:**

["NO or VERY MINOR ethics concerns only"]

**Final Justification:**

I continue to maintain that the paper is a very strong paper on all dimensions. The authors have clarified my questions with appropriate responses along with references. I maintain my recommendation.

**Limitations:**

Yes. Limitations are discussed in the Appendix

**Paper Formatting Concerns:**

None.

**Quality:**

4

**Strengths And Weaknesses:**

- The paper very cleanly discusses the background on the problem in DPO, the authors are trying to solve. The issue with bad probability displacements and squeezing effect is discussed with its right background. The discussion on the valley region of the softmax function and impact on the gradients it has, is quite informative and sets the stage for the upcoming discussion on the proposed solution.

- The paper delves into the mathematical details of the effect of the gap in DPO today. The following discussions are quite useful.
  - The log likelihood decline graph for both the preference and non-preference samples clearly shows the problem with DPO.
  - The discussion on off-policy data collection and its issues with distribution shift.

- The paper has very clear math derivations which makes it quite easy to follow the proposed methodology. Starting from the DPO loss, the paper discusses the characteristics of the loss function and introduces the regularization term with the hyperparameter, which alters the nature of the loss function. Then the paper goes into deriving the gradients of the new loss function. This makes it clear on its intended effect and what the specified change is doing. After that, the paper goes into what's the appropriate value to set for the hyper parameter. It delves into the information theoretical perspective on the labels and the optimal information that we can retrieve from them. And then decides on the appropriate values for the hyperparameter. This presents a very clean and rigorous mathematical analysis of the proposed approach.

- The results section clearly shows the impact of the new method. The evaluations are done on a variety of benchmarks on base/instruct versions of  Qwen2.5-7B and Llama-3.1-8B models. And it seems to show VPO method outperforms on most of the methods against the baselines compared. The details on the experimental setup is also very clearly mentioned and largely makes sense. The plot of log probability change over training on DPO vs VPO makes it clear the effect the approach has, in the squeezing effect.

- Analysis on the effect of the choices of v and graphs in Appendix Figure 8. Is appreciated. It gives a good idea on the effect of the hyper-parameter and how its pushes the approach into different regimes.

- A good set of peripheral information is present in the Appendix, including prompts, additional experiments etc. This leaves very few questions unanswered.

Overall I feel this paper has good mathematical foundations and analysis into the issues faced by DPO, the gaps it causes, how the proposed method solves them along with thorough experiment results which validates them.

I rate the paper 4 in all areas, Quality, Clarity, Significance, Originality.

---

> ### Author Rebuttal · Authors · 2025-07-31
>
> We sincerely appreciate your time and valuable feedback on our research.
>
> ### **Question 1:**
> Thank you for your detailed feedback.
>
> A series of studies [1-4] have confirmed that in reasoning tasks, the log probabilities of preference samples tend to decline, which may even degrade model performance. Specifically, in complex reasoning scenarios, correctness-based preference labeling fails to fully capture the fine-grained quality of model outputs—particularly for questions with definitive answers, where preference labeling often relies solely on answer correctness. Moreover, reasoning chains contain critical tokens that significantly influence the model's reasoning trajectory, yet current labeling paradigms overlook these key elements [2].
>
> These limitations in preference labeling for reasoning tasks may cause DPO to excessively penalize rejected samples containing partially correct reasoning chains during training, leading to probability declines in preference data with similar reasoning chains. [3] also demonstrated that when reasoning preference data is similar, preference optimization failure and probability decline can occur.
>
> On the other hand, longer outputs in reasoning tasks make the distribution of preference sample pairs more un-uniform and harder to analyze in the output space [5], potentially leading to more pronounced "squeezing effect" during model optimization.
>
> ### **Question 2:**
> Thank you for your valuable feedback.
>
> The "valley" region indicates that the model's outputs are trapped in low-probability areas. As analyzed in [6], DPO places greater emphasis on avoiding the generation of non-preferred samples, which causes its log probabilities to decline more rapidly and consequently makes it more prone to falling into the "valley" region.
>
>
>
>
> [1] Pang, Richard Yuanzhe, et al. "Iterative reasoning preference optimization." Advances in Neural Information Processing Systems 37, NIPS 2024.
>
> [2] Lin, Zicheng, et al. "Critical Tokens Matter: Token-Level Contrastive Estimation Enhances LLM's Reasoning Capability."
>
> [3] Pal, Arka, et al. "Smaug: Fixing failure modes of preference optimisation with dpo-positive."
>
> [4] Liu, Zhihan, et al. "Provably mitigating overoptimization in rlhf: Your sft loss is implicitly an adversarial regularizer." Advances in Neural Information Processing Systems 37,NIPS 2024.
>
> [5] Ren, Yi, and Danica J. Sutherland. "Learning dynamics of llm finetuning." The Thirteenth International Conference on Learning Representations. ICLR 2025
>
> [6] Feng, Duanyu, et al. "Towards analyzing and understanding the limitations of dpo: A theoretical perspective, 2024."

---

> > ### Comment · Reviewer_1j6X · 2025-08-03
> >
> > Appreciate the explanation and the references.

---

### Official Review · Reviewer_4dcd · 2025-07-02

**Clarity:** 3
**Significance:** 2
**Originality:** 2
**Rating:** 4
**Confidence:** 3

**Summary:**

The author improved the DPO loss by considering that negative examples might be similar to positive ones, and the negative gradients on negative examples could lead to a decrease in the likelihood of positive examples. To address this, the author proposed an automatic weighting method for the loss on negative examples, leveraging a specific type of v-usable information: p(label|cot) - p(label|X). This measures how much the negative example's chain-of-thought (cot) contributes to predicting the correct answer, thereby reducing the loss weight for high-information negative examples. Experiments on Llama3-8B and Qwen2-7B showed that this method outperforms alignment approaches like DPO and SimPO, as well as the RPO method, which simply increases the likelihood of positive examples.

**Questions:**

1. The use of the terms "preference/non-preference" feels unusual. Why not use "chosen/rejected" or "preferred/non-preferred"?
2. Is maintaining a positive log probability for chosen examples positively correlated with final performance improvement?
3. In Table 1, why do IPO and RPO show significantly different performance across models? Perhaps more careful hyperparameter tuning could improve their results.
4. It would be beneficial if Figure 2 could be expanded to include log probability comparisons with other baselines.

**Ethical Concerns:**

["NO or VERY MINOR ethics concerns only"]

**Final Justification:**

The author's response  has addressed my questions regarding the correlation between v-usable Information and naive similarity metrics. This likely illustrates the distinctiveness of v-usable Information compared to other methods.

Regarding the discussion of RPO and IPO, my initial concern was that the significant performance differences across models might be due to inappropriate hyperparameter choices. The author attributes this to IPO's more conservative constraint, and I can accept this explanation.

The authors did not provide the missing comparative experiments on KL or the curve variations of the baseline in their rebuttal, claiming that these would be included in the revision.

**Limitations:**

Yes

**Quality:**

2

**Strengths And Weaknesses:**

Strengths
1. The method is simple and relatively easy to implement.
2. The author conducted experiments on four different models, with results surpassing the compared baselines.

Weaknesses
1. The current motivation is somewhat heuristic and does not sufficiently justify the necessity of using v-usable information. The experiments lack comparisons between different weighting schemes for negative gradients. For example, how would simpler baselines, such as using the similarity between chosen and rejected samples as the weighting factor (v) perform? What is the correlation between v-usable information and similarity-based metrics? Theoretical support for its necessity would be valuable, for instance, proving whether v-usable information theoretically maximizes the gap.
2. Alignment methods are sensitive to hyperparameters. It would be helpful to plot KL-reward curves to compare different methods under similar KL budgets.

---

> ### Author Rebuttal · Authors · 2025-07-31
>
> ### **Weaknesses 1:**
> We sincerely appreciate your constructive suggestions regarding the weighting scheme, which will further refine our experiments.
>
> In Section 2.3, we investigate the selection principles of different weighting methods, specifically examining how to measure the correlation between Chosen/Rejected pairs in reasoning tasks from both token and information perspectives. Reasoning tasks exhibit prefix similarity and solution path diversity. Since token-based similarity metrics (such as Cosine similarity, Levenshtein edit distance, and Jaccard similarity) are sensitive to variations like word overlap, formal similarity, and surface semantics, they may introduce bias in evaluating the relevance of sample pairs. This makes it difficult for them to accurately assess the similarity between sample pairs in reasoning tasks (particularly when certain reasoning paths contain reflective terms or involve multiple thought processes).
>
> Therefore, we employ V-usable information [1-2] to measure the informational correlation between chosen and rejected responses, as it captures deeper semantic relationships and remains unaffected by surface-level variations. To compare with other similarity-based weighting schemes, we replace the weighting scheme  v with the cosine similarity between Chosen/Rejected pairs. We evaluate the performance of different preference optimization algorithms on Qwen3-14B-Base, keeping the experimental setup consistent with Section 3. The experimental results are presented in the table below.
>
> |   Method   |    MATH    |   MInerva  |   Olympid  |    AIME    |    AMC23   |    GSM8k   |     Avg    |
> |:----------:|:----------:|:----------:|:----------:|:----------:|:----------:|:----------:|:----------:|
> |    Base    |   63.60    |   24.63    |   21.78    |    0.00    |   22.89    |   93.93    |   37.81    |
> |     DPO    |   76.40    |   28.68    |   33.63    |   20.00    |   45.78    |   94.09    |   49.76    |
> |    TDPO    |   70.40    |   26.47    |   27.56    |   13.33    |   38.55    |   94.31    |   45.10    |
> |    Simpo   |   75.60    |   31.25    |   32.44    |   16.67    |   43.37    |   95.75    |   49.18    |
> |     IPO    |   64.60    |   25.00    |   22.81    |   10.00    |   22.89    |   94.24    |   39.92    |
> |     RPO    |   78.00    |   32.35    |   34.67    |   13.33    |   51.81    |   95.68    |   50.97    |
> | **Cosine** | **75.60** | **29.78** | **34.67** | **20.00** | **44.58** | **95.75** | **50.06** |
> |   **VPO**  | **79.00** | **35.66** | **35.41** | **26.67** | **53.01** | **96.06** | **54.30** |
>
> The experimental results and log-probability variation curves demonstrate that V-usable information can more effectively mitigate the "squeezing effect". We will conduct more comprehensive experiments and analysis on the weighting scheme in the revised version.
>
> ### **Weaknesses 2:**
> Thank you for your constructive feedback.
>
> To better understand the differences between these algorithms, we will update the KL curves of various models under different preference optimization methods in the revised version and provide a more detailed analysis.
>
> ### **Questions 1:**
> Thank you for your note regarding term usage.
>
> In preference optimization, the terms "preference/non-preference" and "chosen/rejected" carry equivalent meanings. We will elaborate on and update this terminology in the revised version to ensure clarity.
>
> ### **Questions 2:**
> Thank you for your meaningful feedback.
>
> There is a certain positive correlation between the improvement of model performance and maintaining the probability of chosen examples.  Our analysis in Section 2.2 and the discussion of the "squeezing effect" in [3] demonstrate that  "squeezing effect" causes both chosen and rejected probabilities to decrease simultaneously during DPO training, while the probabilities of certain tokens unrelated to the chosen/rejected pairs become elevated due to this "squeezing".  This phenomenon ultimately leads to unstable model outputs and performance degradation.
>
> Maintaining or even increasing the probability of chosen samples can help alleviate the issue of probability mass shifting to irrelevant tokens during DPO training. Our study achieves this by dynamically constraining negative gradients, which effectively preserves chosen sample probabilities, reduces the generation probability of irrelevant tokens, while maintaining the discriminative power between chosen and rejected samples.
>
> ### **Questions 3:**
> Thank you for your feedback regarding the hyperparameters.
>
> Both IPO and RPO introduce an additional hyperparameter. In the table below, we present the performance variations of IPO and RPO under different hyperparameter settings.
>
> | Method |   Parms   |    MATH    |   MInerva  |   Olympid  |    AIME    |     ACM    |    GSM8k   |     Avg    |
> |:------:|:---------:|:----------:|:----------:|:----------:|:----------:|:----------:|:----------:|:----------:|
> |        |   0.10    |   63.80    |   25.00    |   22.07    |    3.33    |   28.92    |   94.31    |   39.57    |
> |   IPO  |   0.50    |   64.60    |   26.47    |   23.11    |    3.33    |   26.51    |   94.16    |   39.70    |
> |        |   1.00    |   63.40    |   22.06    |   22.81    |    3.33    |   22.89    |   93.78    |   38.05    |
> |        |**2.00** |**64.60** |**25.00** |**22.81** |**10.00** |**22.89** |**94.24** |**39.92** |
> |        |   0.25    |   75.60    |   31.41    |   32.74    |   13.33    |   42.17    |   96.06    |   48.55    |
> |   RPO  |   0.50    |   75.00    |   33.48    |   33.78    |   10.00    |   36.14    |   95.98    |   47.40    |
> |        |**1.00** |**78.00** |**32.35** |**34.67** |**13.33** |**51.81** |**95.68** |**50.97** |
> |        |   2.00    |   77.00    |   33.48    |   33.93    |   13.33    |   45.78    |   95.75    |   49.88    |
>
>
> ### **Question 4:**
> Thank you for your valuable feedback on the analysis of our experimental results, which will make our experimental analysis more comprehensive. We will update the log-probability comparison curves with other baselines in the revised version.
>
>
>
>
> [1] Kawin Ethayarajh, Yejin Choi, and Swabha Swayamdipta. Understanding dataset difficulty with V-usable information, The Thirty-Ninth International Conference on Machine Learning, ICML 2022.
>
> [2] Yilun Xu, Shengjia Zhao, Jiaming Song, Russell Stewart, and Stefano Ermon. A theory of usable information under computational constraints. The Eighth International Conference on Learning Representations ICLR 2020.
>
> [3] Ren, Yi, and Danica J. Sutherland. "Learning dynamics of llm finetuning." The Thirteenth International Conference on Learning Representations. ICLR 2025

---

> > ### Comment · Reviewer_4dcd · 2025-08-06
> >
> > Thank you for the author's reply.
> >
> > Regarding the issue of different measures, I agree that intuitively V-usable information might be more reasonable than other similarity measures.
> >
> > I suggest the authors try calculating correlation metrics, such as Spearman correlation, between V-usable information and, for example, embedding similarity or text similarity, to illustrate the extent of their similarities and differences.
> >
> > I've noticed that V-usable information is also dynamic compared to these other measures. How significant is the difference between dynamic V-usable information and preprocessed V-usable information?
> >
> > As for the gap between IPO and RPO, what I’m particularly curious about is why, in the experimental results table, RPO performs significantly better than IPO for some base models, while the opposite is true for others. Why does this phenomenon occur?

---

> > > ### Author Response · Authors · 2025-08-07
> > >
> > > ### **Question 1:**
> > >
> > > Thank you for your constructive feedback.
> > >
> > > We have calculated the Spearman correlation between V-usable information, Embedding cosine similarity, and text-based Jaccard similarity. The results are as follows:
> > >
> > > | Factor | Spearman's rank correlation | |
> > > |:-----------------------------:|:---------------------------:|:-------:|
> > > | | correlation | p_value |
> > > | V-usbale information & Cosine | 0.01 | 0.45 |
> > > | V-usbale information& Jaccard | -0.01 | 0.23 |
> > > | Cosine & Jaccard | 0.46 | 0.01 |
> > >
> > >
> > > The Spearman correlation results demonstrate statistically significant differences between V-usable information and both cosine similarity and Jaccard similarity.
> > > We further employed Jaccard similarity as a factor for preference optimization, with experimental results (Qwen3-14B-Base) presented in the following table:
> > >
> > >
> > > | Factor | MATH | MInerva | Olympid | AIME | AMC23| GSM8k | Avg |
> > > |:------------------------:|:------:|:---------:|:---------:|:---------:|:---------:|:---------:|:--------:|
> > > | Cosine | 75.60 |29.78| 34.67 | 20.00 | 44.58 | 95.75 | 50.06 |
> > > | Jaccard| 78.00| 32.35| **37.78** | 16.67 | 49.40 | 95.98 | 51.70 |
> > > | **V-usbale information** | **79.00** | **35.66** | 35.41 | **26.67** | **53.01** | **96.06** | **54.30** |
> > >
> > > This ablation study demonstrates that using V-usable information as a factor for negative gradient constraints proves more effective compared to other similarity-based methods.
> > >
> > > We infer that the advantage of Jaccard similarity over cosine similarity may stem from its accidental alignment with a more optimal hyperparameter space rather than superiority intrinsic capability in assessing the correlation of chosen/rejected responses, since their Spearman correlation remains persistently high (correlation=0.46, p-value=0.01) and cosine similarity demonstrates greater semantic capture capacity in most reasoning tasks.
> > >
> > > In contrast, the results in Table 2 (main text) demonstrate that while optimal constraint hyperparameters vary across different models, VPO consistently achieves superior performance. We will update the results of different factors under other series of models in the revised version.
> > >
> > > In summary, compared to other embedding-based and text similarity metrics, V-usable information captures deeper linguistic structures and semantic relationships, providing explanatory insights into causal relationships and dependency directions. Leveraging these advantages, VPO delivers more significant performance improvements.
> > >
> > > ### **Question 2:**
> > >
> > > Thank you for your response.
> > >
> > > The V-usable information used in our study was all pre-computed. We did not use dynamically V-usable information from the policy model during training, because using V-usable information calculated with updated policy models might increase the divergence between the policy model and reference model, leading to more severe "squeezing effect".
> > >
> > > ### **Question 3:**
> > >
> > > Thank you for your careful observation and suggestions.
> > >
> > > In fact, the key improvement of IPO lies in its mitigation of the issue where DPO converges to a deterministic policy, leading to a significant divergence between the policy model and the reference model due to the weakening of KL regularization. IPO addresses this by introducing a regularization parameter to control the proximity between the policy model and the reference model, typically maintains closer alignment with the reference model.
> > > Our experimental results are consistent with the findings in the IPO paper. We observed that IPO achieves lower KL reward and exhibits smaller fluctuations in logp during training compared to DPO. Additionally, [1] suggests that IPO's regularization helps alleviate the "squeezing effect" to some extent, thereby maintaining model performance even when the "squeezing effect" is pronounced.
> > >
> > > For RPO, unlike DPO, it incorporates an NLL loss. This NLL loss helps RPO maintain the logp of chosen samples to some extent, but it still fails to avoid the "squeezing effect". In instruct models where the "squeezing effect" is more pronounced, we observe performance degradation with this algorithm.
> > >
> > >
> > > In conclusion, when the "squeezing effect" is weak, most algorithms demonstrate performance improvements. However, IPO exhibits relatively smaller gains due to its incorporated regularization parameters, resulting in inferior performance compared to RPO. In contrast, when the "squeezing effect" becomes strong, RPO's effectiveness diminishes to some extent while IPO maintains stable performance, ultimately leading to superior results over RPO.
> > >
> > > We express our respectful gratitude to the reviewer for your time and effort. We hope the various additional experiments could strengthen our paper and we will revise the paper according to your valuable feedback. If you have any remaining questions or think some experiments are insufficient, feel free to respond to us at your convenience.
> > >
> > > [1] Ren, Yi, and Danica J. Sutherland. "Learning dynamics of llm finetuning.", ICLR 2025

---

> > > > ### Comment · Reviewer_4dcd · 2025-08-08
> > > >
> > > > Thank you for the author's response, which has addressed my questions regarding the correlation between v-usable Information and naive similarity metrics. This likely illustrates the distinctiveness of v-usable Information compared to other methods.
> > > >
> > > > Regarding the discussion of RPO and IPO, my initial concern was that the significant performance differences across models might be due to inappropriate hyperparameter choices. The author attributes this to IPO's more conservative constraint, and I can accept this explanation.
> > > >
> > > > I will increase my score.
> > > >
> > > > Additionally, I’d like to ask: since these metrics show very weak correlation with v-usable information, apart from these naive similarity metrics, could there be other metrics that exhibit a slightly stronger correlation with v-usable information?

---

> ### Author Response · Authors · 2025-08-09
>
> We are very pleased that our rebuttal has addressed your concerns. We also greatly appreciate your constructive feedback.
>
>
> V-usable information, as an extension of mutual information oriented toward the effectiveness of prediction tasks, has been proven to be more effective than mutual information in representation learning [1]. A series of extended metrics for mutual information (such as MINE [2] and Mixed-KSG [3]) exhibit a positive correlation with V-usable information, as shown in Figure 1 of [1].
>
>
> In addition to these mutual information extension methods, we believe that information-theoretic distribution similarity measures closely related to mutual information—particularly KL divergence and cross-entropy, may also exhibit strong positive correlations with V-usable information. However, their application is limited by computational challenges arising from length mismatches when comparing variable-length sequences.
>
>
> Model-based similarity metrics using internal states—such as those based on hidden states or attention scores—can capture similarity through learned representations, though this requires significant computational overhead. Additionally, we note several similarity optimization methods based on LLM outputs [4-6], which transform semantic textual similarity into a text generation problem. In future work, we will explore the correlation between these methods and V-usable information, as well as their performance when used as factors for preference optimization.
>
> [1] Xu, Yilun, et al. "A theory of usable information under computational constraints." ICLR 2020.
>
> [2] Ishmael Belghazi, Mohamed, et al. "MINE: mutual information neural estimation."  ICML 2018.
>
> [3] Gao, Weihao, et al. "Estimating mutual information for discrete-continuous mixtures."  NIPS 2017.
>
> [4] Ravfogel, Shauli, et al. "Description-based text similarity." COLM 2024.
>
> [5] Gatto, Joseph, et al. "Text encoders lack knowledge: Leveraging generative llms for domain-specific semantic textual similarity." EMNLP2023.
>
> [6] Xu, Shaochen, et al. "Reasoning before comparison: LLM-enhanced semantic similarity metrics for domain specialized text analysis."

---

### Official Review · Reviewer_QVPm · 2025-07-03

**Clarity:** 3
**Significance:** 3
**Originality:** 3
**Rating:** 4
**Confidence:** 3

**Summary:**

The paper proposes VPO, a new preference optimization method that improves on Direct Preference Optimization (DPO) for reasoning tasks. VPO tackles the squeezing effect where DPO’s negative gradients can harm both preference and non-preference confidence. By selectively constraining negative gradients based on V-usable information, VPO better preserves the distinction between correct and incorrect outputs. Experiments on various mathematical reasoning benchmarks show consistent performance gains over DPO and its recent variants.

**Questions:**

Is the extra computation for measuring V-usable information significant at large scale?

**Ethical Concerns:**

["NO or VERY MINOR ethics concerns only"]

**Final Justification:**

This is a good paper with solid results. The authors further provide detailed rebuttals that addressed my concerns. Thus, I will keep my rating.

**Limitations:**

VPO assumes clear reasoning chains and usable intermediate steps, which might not exist in all tasks. It also depends on the quality of the V-usable information estimate, which could be noisy for less structured data.

**Quality:**

3

**Strengths And Weaknesses:**

Strengths
The paper addresses a clear, well-motivated weakness of DPO for reasoning-heavy tasks. Its key idea — using V-usable information to adaptively constrain gradients — is both theoretically sound and practically demonstrated. Strong empirical results, with thorough ablations, show clear improvements across multiple datasets and models.

Weaknesses
The approach focuses mainly on mathematical reasoning; generalization to more open-ended or multi-domain tasks is not fully shown. The method’s reliance on precomputed reasoning chains may limit applicability for tasks with less structured intermediate steps.

---

> ### Author Rebuttal · Authors · 2025-07-31
>
> ### **Weaknesses:**
> We thank you for the constructive feedback about this paper.
>
> In Figure 3, we present a performance comparison between VPO and other preference optimization algorithms on ARC-C, which includes: Definition, Basic Facts & Properties, Structure, Processes & Causal, Teleology / Purpose, among others. Some of these domains involve less structured intermediate steps in reasoning (Problems in Definition-type tasks, such as "What is a worldwide increase in temperature called?"). The results in Figure 3 demonstrate that VPO maintains a strong performance improvement even on commonsense tasks beyond mathematical tasks.
>
>
> Additionally, we perform preference optimization on the Qwen3-14B model using the ScienceQA training set. ScienceQA spans 12 grade levels and features diverse input sources, covering multi-domain data such as History, Writing Strategies, Vocabulary, Biology, Civics, Physics, and Geography. For each question, we sample 5 responses, constructing a total of 7K preference sample pairs for Qwen3-14B. The experimental results on the ScienceQA test set are shown in the table below.
>
> |  Method |  Scienceqa |
> |:-------:|:----------:|
> |   Base  |   74.70    |
> |   DPO   |   72.92    |
> |   TDPO  |   74.70    |
> |  SimPO  |   76.49    |
> |   IPO   |   75.30    |
> |   RPO   |   73.81    |
> | **VPO** | **77.68** |
>
> The above results demonstrate that VPO can still achieve significant performance improvements across multi-domain tasks—even those with less structured intermediate steps. We will include additional experiments and analyses on these tasks in the revised version.
>
> ### **Questions:**
> Thank you for your valuable feedback regarding the computational cost of V-usable information.
>
> In fact, computing V-usable information only requires a single forward pass on the rejected sample  $\theta(y|x,c_l)$  and $\theta(y|x)$ . To further reduce computational overhead, we can concatenate the Reject sample $(x,c_l, y)$ with the Question-Answer sample $(x,y)$ for joint processing. All V-usable information calculations in the paper can be completed within 30 minutes on a single H20 GPU. Additionally, several proxy methods [1] (capable of achieving up to 30x speedup) have been proposed and could be extended to compute V-usable information, enabling more efficient large-scale applications. We will continue to explore optimizations for reducing the computational overhead of V-usable information in future work.
> ### **Limitations:**
> Thank you for your constructive feedback on scalability.
>
> Our experiments on ARC-C and ScienceQA investigate VPO's performance across multi-domain tasks with less structured reasoning chains. Experimental results validate VPO's scalability on general tasks. We will further investigate VPO's performance on less structured data in future research.
>
> [1] Liu, Fengyuan, Nikhil Kandpal, and Colin Raffel. "AttriBoT: A Bag of Tricks for Efficiently Approximating Leave-One-Out Context Attribution." The Thirteenth International Conference on Learning Representations. ICLR 2025

---

> > ### Comment · Reviewer_QVPm · 2025-08-06
> >
> > Thanks for the detailed response and experimental results. I will keep my rating for this paper.

---

### Official Review · Reviewer_Xe5e · 2025-07-05

**Clarity:** 3
**Significance:** 3
**Originality:** 3
**Rating:** 5
**Confidence:** 4

**Summary:**

This paper introduces VPO, a novel negative gradient constraint method designed to address issues in DPO when applied to reasoning tasks. The core problem identified is that DPO, especially with a softmax output head, can lead to a "squeezing effect" where the confidence of both preference and non-preference samples decreases, and confidence in unrelated tokens increases. VPO aims to alleviate this by using "V-usable information" to measure the similarity between preference pairs and selectively constrain negative gradients for human non-preference samples. The authors have shown the effectiveness of VPO over baseline methods across different reasoning benchmarks.

**Questions:**

See weakness and an additional question as below.

1. Did the authorsd evaluate VPO on AIME?

**Ethical Concerns:**

["NO or VERY MINOR ethics concerns only"]

**Limitations:**

yes

**Quality:**

3

**Strengths And Weaknesses:**

Strengths
1. The paper highlights a specific problem with DPO in reasoning tasks, where it can inadvertently reduce confidence in relevant samples and increase confidence in irrelevant tokens due to large negative gradients on low-confidence samples.
2. The concept of V-usable information is proposed as a metric to quantify the similarity between preference pairs, which is then used to guide the negative gradient constraint. It is technically simple and theoretically sound.
3. The paper provides experimental results demonstrating VPO's effectiveness across various reasoning tasks showing improvements in accuracy and win rates compared to DPO and some of its recent variants.

Weaknesses
1. The paper explicitly focuses on reasoning tasks. It is unclear how VPO would perform or if the "squeezing effect" is as prominent in other types of tasks (e.g., creative writing, summarization). A brief discussion on the potential applicability or limitations in non-reasoning contexts would be valuable.
2. The experiments were conducted with Llama-3.1-8B and Qwen-2.5-7B. It will be good to see how VPO performs and generalizes to larger models and/or more powerful models (e.g., the latest 14B / 32B Qwen-3).

---

> ### Author Rebuttal · Authors · 2025-07-31
>
> ### **Weaknesses 1:**
> We thank you for the insightful comments and constructive feedback about this paper.
>
> The study in [1] performed off-policy DPO training on the Pythia/Qwen1.5 model using the UltraFeedback and Anthropic-HH datasets. These datasets were specifically curated to enhance the model's performance in instruction following, truthfulness, honesty, and helpfulness, encompassing several relevant task categories including: World Knowledge Question Answering, Creative Writing and Content Generation,Document-based Assistance. Experimental results (Figure 4) of [1] demonstrate significant "squeezing effect"  across these tasks.
>
> Our new experiments with Qwen3-14B using UltraFeedback data revealed consistent behavioral patterns. Overall, the "squeezing effect" of DPO on UltraFeedback tasks show no significant difference compared to reasoning tasks (we will provide further discussion in the revised version).
> Due to the absence of ground-truth annotations in the UltraFeedback dataset,  calculation of V-usable information remains infeasible. In future work, we will conduct an in-depth investigation of mitigation strategies for "squeezing effect" in non-reasoning scenarios.
>
> ### **Weaknesses 2:**
> Thank you for your valuable feedback.
>
> We have conducted a comprehensive comparison of various preference optimization methods on the Qwen3-14B-Base model, maintaining identical experimental settings to those described in Section 3 of our paper. The results are presented in the following table.
>
> |  Method |    MATH    |   MInerva  |   Olympid  |    AIME    |    AMC23   |    GSM8k   |     Avg    |
> |:-------:|:----------:|:----------:|:----------:|:----------:|:----------:|:----------:|:----------:|
> |   Base  |   63.60    |   24.63    |   21.78    |    0.00    |   22.89    |   93.93    |   37.81    |
> |   DPO   |   76.40    |   28.68    |   33.63    |   20.00    |   45.78    |   94.09    |   49.76    |
> |   TDPO  |   70.40    |   26.47    |   27.56    |   13.33    |   38.55    |   94.31    |   45.10    |
> |  Simpo  |   75.60    |   31.25    |   32.44    |   16.67    |   43.37    |   95.75    |   49.18    |
> |   IPO   |   64.60    |   25.00    |   22.81    |   10.00    |   22.89    |   94.24    |   39.92    |
> |   RPO   |   78.00    |   32.35    |   34.67    |   13.33    |   51.81    |   95.68    |   50.97    |
> | **VPO** | **79.00** | **35.66** | **35.41** | **26.67** | **53.01** | **96.06** | **54.30** |
>
> The experimental results demonstrate that VPO can maintain good performance improvements on larger models.
>
> ### **Questions 1:**
> Thank you for your valuable feedback.
>
>  We did not evaluate Qwen 2.5 and LLaMA-3.1-8B on the AIME benchmark due to the limited problem set in AIME24 and the fact that the baseline LLM, LLaMA-3.1-8B model without additional fine-tuning struggles to achieve meaningful performance improvements on AIME tasks.
> The experimental results from our Qwen3-14B evaluation on AIME (presented in the preceding table) confirm that VPO outperforms other preference optimization approaches.
>
> [1] Ren, Yi, and Danica J. Sutherland. "Learning dynamics of llm finetuning." The Thirteenth International Conference on Learning Representations. ICLR 2025

---

### Note · Authors · 2025-08-13

We sincerely appreciate the effort of all reviewers and AC in reviewing our paper. We are deeply grateful for the insightful feedback provided by the reviewers that strengthened our work. Below, we summarize our paper and the concerns that were addressed during the rebuttal period.

**The core contribution of VPO lies in:**

**1. Negative Gradient Constraint of DPO.** We constrain the DPO's negative gradient to reduce the "squeezing effect," improving the positive gradient of chosen samples and enhancing consistency with the log-likelihood optimization objective in inference. Table 2 shows that DPO with this constraint outperforms the original DPO on the Llama and Qwen series models in almost all hyperparameter settings.

**2. Selective Negative Gradient Constraint Based on V-usable Information.** Using V-usable information to measure the similarity between Chosen and Rejected samples, VPO selectively constrains the negative gradient of Rejected samples. This balances alleviating the "squeezing effect" in DPO with maintaining effective preference learning. Experiments show that, on Qwen2.5-7B-Base, VPO achieves 7.80% and 13.25% improvement over DPO on MATH500 and AMC23, respectively.

**Addressed Concerns:**

**1. Analysis:** The "squeezing effect" of DPO on open-ended tasks was analyzed based on Ultrafeedback (Xe5e W1); We also analyzed the correlation between the positivity of DPO chosen samples and performance improvement (4dcd Q2), as well as the causes of the enhanced "squeezing effect" and rejected sample logp falling into the "valley" region in inference tasks (1j6X Q1, Q2).

**2. Cost:** The feasibility of calculating V-usable information in large-scale experiments was analyzed (QVPm Q1).

**3. Generalization:** We include the performance of Qwen3-14B-Base on mathematical reasoning tasks and the AIME24 test (Xe5e W2 and Q1). We also add a comparison of VPO with other methods on multi-domain, less-structured tasks (QVPm W1, L1).

**4. Ablation:** We compare the V-usable information based negative gradient constraint method with other similarity-based methods and analyze the correlations between these metrics (4dcd W1).

During the discussion, we address all the concerns raised by the reviewers, and we will carefully revise the manuscript based on the discussion.

We sincerely hope that you will take these factors into account when making your decision. Thank you again for your time and effort.

Best regards,

Submission 18650 Authors.

---

### Decision · Program_Chairs · 2025-09-17

**Decision:**

Accept (spotlight)

**Comment:**

This paper presents VPO as a thoughtful extension of DPO. It squarely addresses the “squeezing effect” that harms reasoning performance. I appreciate how the authors both introduced the negative gradient constraint and also backed it up with careful theoretical discussion. Also, with extensive experiments across multiple models and benchmarks.

The reviewers’ concerns about generalization and computational cost were addressed with additional experiments. While some questions remain about broader applicability beyond reasoning-heavy settings, I see this as a strong and technically solid contribution worth accepting.